# Needs and Quality of Life of Caregivers of Patients with Prolonged Disorders of Consciousness

**DOI:** 10.3390/brainsci13020308

**Published:** 2023-02-11

**Authors:** Olivia Gosseries, Caroline Schnakers, Audrey Vanhaudenhuyse, Charlotte Martial, Charlène Aubinet, Vanessa Charland-Verville, Aurore Thibaut, Jitka Annen, Didier Ledoux, Steven Laureys, Charlotte Grégoire

**Affiliations:** 1Coma Science Group, GIGA-Consciousness, University of Liège, Avenue de l’Hôpital, 1, 4000 Liège, Belgium; 2Centre du Cerveau, University Hospital of Liège, 4000 Liège, Belgium; 3Sensation and Perception Research Group, GIGA-Consciousness, University of Liège, 4000 Liège, Belgium; 4Research Institute, Casa Colina Hospital and Centers for Healthcare, Pomona, CA 91767, USA; 5Interdisciplinary Algology Center, University Hospital of Liège, 4000 Liège, Belgium; 6Psychology and Neuroscience of Cognition Research Unit, University of Liège, 4000 Liège, Belgium; 7GIGA-Consciousness, Coma Science Group & Neurology Department, University and CHU of Liège, 4000 Liège, Belgium; 8Department of Intensive Care, University Hospital of Liège, 4000 Liège, Belgium; 9CERVO Research Center, Laval University, Québec, QC G1E 1T2, Canada

**Keywords:** brain injury, family needs, caregiver, vegetative state, unresponsive wakefulness syndrome, minimally conscious state, disorders of consciousness, quality of life, distress

## Abstract

**Background**. Many patients with severe brain damage may survive and remain in a prolonged disorder of consciousness (PDoC), impacting the quality of life (QoL) and needs of their family caregivers. However, the current literature on the factors influencing these needs is contradictory. We aim to describe the needs, QoL, and emotional distress of caregivers of patients with PDoC. **Methods**. Questionnaires investigating the importance and satisfaction of six categories of needs (i.e., health information, emotional, instrumental, and professional supports, community support network, and involvement in care), QoL, and emotional distress were completed by the main caregivers of PDoC patients. **Results**. We analyzed 177 questionnaires. Seventy-nine percent of the needs were considered as important or very important, and 44% were partially met or unmet. The needs for health information and professional support were the most important, while the needs for involvement in care and for health information were the most satisfied. Mean QoL was low and emotional distress high. Variables such as care setting and time since brain injury affected the level of QoL and distress. **Conclusion**. The needs for health information and professional support should receive particular attention. Given their low QoL and high distress, adequate support structures should be provided to caregivers of PDoC patients.

## 1. Introduction

### 1.1. Impact of Prolonged Disorders of Consciousness on Caregivers

Due to recent progress in intensive care, an increasing number of patients with severe brain injury recover from coma, defined by an absence of arousal and awareness that lasts for a minimum of one hour and up to several days or weeks [1]. Coma may evolve towards an unresponsive wakefulness syndrome/vegetative state (UWS/VS), characterized by the recovery of arousal but without any signs of awareness [2,3]. Patients may subsequently recover partial consciousness (i.e., minimally conscious state—MCS) with inconsistent but reproducible goal-directed behaviors, and further emerge from the MCS (EMCS) when functional communication or object use is re-established [4,5]. Some patients can also evolve to a locked-in syndrome (LIS), characterized by a quadriplegia or quadriparesis with preserved consciousness, cognitive abilities, and sensory pathways [6,7]. Coma, UWS, and MCS are often referred to under the umbrella of “disorders of consciousness” (DoC). When lasting more than 28 days, the term “prolonged disorder of consciousness” (PDoC) is used. This condition can last for months or even decades and generally involves prolonged assistance from healthcare professionals and informal caregivers, mostly family members. As a higher number of patients live longer, at home or in long-term care facilities, the burden of the family caregivers increases [8,9,10,11]. Only a few studies focused on family caregivers of patients with severe brain injury [10], and all of them reported many physical (e.g., pain, sleep disturbances, fatigue, general health), psychological (mainly depression and anxiety, but also burnout, impaired cognitive abilities, guilt, dysfunctional coping strategies, and prolonged grief disorder), social (e.g., personal relationships, social support), and environmental (e.g., financial resources, leisure activity) difficulties, leading to a high number of unmet needs (e.g., social, emotional or financial supports, information about their relative’s health, consideration from the medical team) [8,10,11,12,13,14,15,16,17]. Importantly, the need for medical information is often considered as one of the most important, if not the most important, by the caregivers [15,18]. However, it is not always fully satisfied, as interactions with the healthcare system are often described as an additional burden. Indeed, several studies highlighted the caregivers’ unsatisfied needs regarding the flexibility and coordination of care pathways, the access to medical teams and information, the involvement in decisions about their relative’s care, and the access to care resources (e.g., rehabilitation program, physical therapy) or support services [8,10,15,18,19].

Some characteristics inherently linked to the caregiver and the patient seem to influence the caregiver’s quality of life (QoL) and needs, but their impact is often unclear, as detailed below. Indeed, these studies used different methodologies (e.g., various questionnaires, qualitative interviews), and focused on the caregivers of patients with different diagnoses, care settings, or time elapse since brain injury. Table 1 summaries the contradictory findings of the studies discussed below.

### 1.2. Impact of the Caregiver’s Profile

Regarding the caregivers, different studies showed that female caregivers report more depression, anxiety, and prolonged grief disorders than men [9,16,17,20]. However, another study showed no difference between male and female caregivers regarding burden [21]. In any case, the presence of depression or anxiety [8,14,17] and a high number of unmet needs [15] are known to be associated with a poor caregiver’s QoL. In addition, some studies showed that the caregiver’s burden differed according to their relationship with the patient, with spouses reporting higher burden and anxiety [16,22], while another study showed no difference at this level [12]. The impact of the caregiver’s age is also unclear. For example, some studies reported a link between a young age and prolonged grief disorder [17], while others showed no link between age and depression [17].

### 1.3. Impact of the Patient’s Profile

Regarding the patients, the time since the brain injury seems to be associated with different strains, with more social support reported by the caregiver shortly after the brain injury, and financial, social, and marital negative consequences arising over time [10]. Studies are however contradictory concerning the evolution of the caregiver’s burden over time. More specifically, two reviews showed that QoL, number of needs, and social support decreased over time [8,17], but the evolution of psychological distress and burden is less clear. Some studies showed that they remain constant [17], decrease [16], or increase [17] over time, while another argued that there is no link at all between the time of the brain injury and the caregiver’s burden [13]. Concerning the influence of the patient’s level of consciousness, a study showed higher needs for help and supportive assistance [23] as well as for emotional and social support [13] in cases of MCS compared to UWS patients. Another study showed that a higher level of consciousness was linked to less anxiety in the caregiver [16]. However, other studies showed no difference in caregiver burden, psychological symptoms, and needs in relation to the patient’s diagnosis [13,17,24]. Regarding the care setting of the patient (i.e., at home or in specialized care facilities), studies are also contradictory. Indeed, a review showed increased or decreased anxiety in caregivers whose relative was at home compared to those whose relative was in a specialized care facility, depending on the studies considered [11], while two other studies showed no difference in burden, distress and grief according to the care setting [17,25]. Some authors even argue that this factor would not have any influence on the caregiver burden [14]. Finally, the age of the patient could also impact the caregivers’ QoL, as some studies showed an increased likelihood of prolonged grief disorder and higher anxiety when the patient was younger [16,17].

### 1.4. Objectives

Given the serious negative impact of caregiving on QoL of relatives of patients with PDoC, understanding who the caregivers with particularly low QoL and high distress are seems essential. Investigating their needs along with their QoL would also bring useful information and allow for development of more relevant support structures [8,9,19]. In this context, the first aim of this study is to assess the needs (in terms of importance and satisfaction, see Section 2.2. Assessments for the categories of needs) of caregivers of patients with severe brain injury, in parallel to their QoL, psychological distress, and opinions about end-of-life decisions. We hypothesize that the need for health information would be the most important need and one of the least satisfied. We also assume that the caregivers’ QoL would be low, and that most of them would report depressive thoughts and anxiety. Our second aim is to investigate the differences between the caregivers according to their level of QoL and psychological distress (i.e., levels of anxiety and depressive thoughts), in terms of needs’ importance and satisfaction, psychological distress, socio-demographic and patient-related medical variables. We hypothesize that caregivers with a low QoL would also report more unmet needs, more anxiety, and more depressive thoughts, and that those with higher anxiety and depressive thoughts would also have a higher number of unmet needs. No a priori hypotheses are formulated regarding the differences in terms of age, gender, patients’ level of consciousness, and care setting, as previous studies did not reach a consensus. Finally, we aim to explore the differences in needs’ importance and satisfaction, QoL, psychological distress, and end-of-life decisions according to different socio-demographic and medical variables (i.e., caregiver’s gender, relationship with the patient, patient’s level of consciousness, care setting, age of the caregiver and the patient, and time since brain injury). As the scientific literature on these factors is quite contradictory, we do not formulate any a priori hypothesis.

## 2. Methods

### 2.1. Recruitment

The main caregivers of patients with severe brain injuries anonymously participated in this cross-sectional study. Inclusion criteria were as follows: (i) the participant was the main caregiver (i.e., person being responsible for the patient’s care) of a patient suffering from severe brain injuries with a diagnosis of coma, UWS, MCS, EMCS or LIS, and (ii) the patient was living at home or hospitalized in one of the subacute or long-term care facilities involved in the study [26], mostly in Belgium but also outside Belgium (N = 32; France, Holland, Germany, Luxembourg, Italy, United Kingdom or Greece). The questionnaires and a form explaining the aim of the study were sent by regular mail to a contact person employed in each facility, or to some caregivers our team knew and who had their relative at home. Participants were explicitly asked to anonymously fill in the questionnaires, only if they wanted to participate in the study. Participants’ consent was implied by submitting the completed questionnaires. The completed questionnaires were then returned by the contact person to our research team. The study was approved by the Ethical Committee of the Faculty of Medicine of the University of Liège.

### 2.2. Assessments

Each participant completed a battery of four self-reported questionnaires composed of:

Socio-demographic and medical questionnaire: information was collected about the caregiver (i.e., age, gender, place of living, relationship to the patient, profession), the patient (i.e., age, time elapsed since brain injury, level of consciousness [coma, UWS, MCS, EMCS, LIS]), and the care setting (i.e., neuro-rehabilitation center, nursing home, general hospital, or home). Regarding the patients’ level of consciousness, no standardized assessment had been conducted, and this information was given by the caregiver. Two questions also concerned end-of-life decision (i.e., never considered, considered for a while but not anymore, desired), and the continuation of therapy (i.e., no limit, do not reanimate, do not add any therapy nor extend the ongoing therapy, progressive discontinuation of therapy).

Family Needs Questionnaire (FNQ) [27]: this questionnaire has been used to assess the needs of relatives of patients suffering from severe brain injury [15,18,28]. The questionnaire is composed of 40 items, including needs that can be categorized in 6 domains: health information (e.g., “I need to be shown that medical, educational or rehabilitation staff respect the patient’s needs or wishes”), emotional (e.g., “I need help getting over my doubts and fears about the future”), instrumental (e.g., “I need to have help keeping the house (e.g., shopping, cleaning, cooking, etc.)”), and professional supports (e.g., “I need to be shown what to do when the patient is upset or acting strange”), as well as community support network (e.g., “I need to have a professional to turn to for advice or services when the patient needs help”), and involvement in care (e.g., “I need to be told daily what is being done with or for the patient”). Respondents were asked to estimate the importance of each need on a Likert scale ranging from 0 (not important at all) to 3 (very important), and to indicate their satisfaction for each need (i.e., met (=2), partially met (=1), or unmet (=0)). In this study, we excluded one item of the questionnaire (i.e., “to have complete information on drug or alcohol problems and treatment”) as it was not relevant for our population. Different scores were calculated based on the remaining 39 items: (1) the number of met, partially met, and unmet needs, (2) a mean importance score, and (3) a mean satisfaction score for each category of needs, expressed in percentages. To do so, the importance and satisfaction scores of each item of each category of needs were added to obtain two total scores for each category of needs: importance and satisfaction. These sums were transformed into percentages according to the maximum score possible for each dimension, with 100% expressing the highest importance or satisfaction for this category of needs, and 0% the lowest importance or satisfaction. Person mean imputation was used to deal with the missing data in this questionnaire, consisting of calculating the average over the available items and multiplying that with the number of items in the questionnaire [29].

The Anamnestic Comparative Self-Assessment (ACSA) scale [30]: this scale assesses the QoL, taking as endpoints the worst and the best time of the participant’s life, and assigning a score of −5 and +5, respectively, to these moments. Respondents were asked to compare their current life situation (i.e., the last two weeks) with those two moments, and to assign a score between −5 and +5.

Psychological distress: caregivers were invited to report their level of anxiety (i.e., absent, moderate, or extreme) and the presence of depressive thoughts (i.e., never, frequently or often present).

### 2.3. Statistical Analyses

All statistical analyses were performed using Statistica 13.3 (TIBCO Software Inc., Palo Alto, USA) and SPSS Statistics 25 (IBM). When looking at the FNQ, ACSA, and questions related to psychological distress, we used descriptive statistics such as frequencies and percentages for categorical variables, as well as means and SD for continuous variables. Regarding our second and third aims (i.e., (1) characteristics of the caregivers who have a low vs. high QoL and psychological distress, and (2) differences between the caregivers when considering other socio-demographic and medical variables), we used Chi-square (for categorical variables) and Kruskal–Wallis (for continuous variables) tests, as well as Spearman’s rank order correlation. As suggested by Cohen [31], the following criteria were used for the interpretation of the correlation coefficients: 0.00–0.10 = trivial; 0.10–0.30 = small; 0.30–0.50 = moderate; >0.50 = large. All tests were two-tailed and a *p* < 0.05 was used to determine statistical significance.

## 3. Results

### 3.1. Description of the Sample

#### 3.1.1. Socio-Demographic Variables

Out of the 215 questionnaires sent out, 177 completed questionnaires were returned (82.3%). The socio-demographic characteristics of the caregivers and socio-demographic and medical characteristics of the patients are shown in Table 2. Caregivers were on average 52 ± 13 years old and were mostly women (57%). Most were the patient’s parents or spouse (80%), and less frequently the child (9%) or the sibling (9%). The patients were on average 43 ± 15 years old and were mostly hospitalized in a neuro-rehabilitation center or a nursing home (75%). Time since brain injury was extremely variable and ranged from 1 month to nearly 30 years. Most patients were considered as being in a MCS (50%) or in a UWS (14%).

#### 3.1.2. Caregivers’ Needs

When considering the whole sample, the mean numbers of important and very important needs were 7.28/39 (SD = 5.70) and 23.45/39 (SD = 8.97), respectively, accounting for 78.8% of all the needs investigated. Regarding satisfaction, the mean numbers of unmet and partially met needs were 7.29/39 (SD = 6.73) and 9.97/39 (SD = 6.97), respectively, representing together 44.3% of the needs investigated. Table 3 details the mean importance and satisfaction for the six categories of needs. The most important category was the need for health information (95%). Two of the most important categories of needs (i.e., need for health information and need for involvement with care) were also the two most satisfied (73% and 74%, respectively). The least satisfied category of needs was related to emotional support, which was also considered the least important. Figure 1 and Figure 2 detail the number of needs according to their importance and satisfaction, and the mean importance and satisfaction of each category of needs, respectively.

#### 3.1.3. Caregivers’ QoL and Opinions about End-of-Life Decisions

Table 4 details the caregivers’ anxious and depressive symptoms, QoL, and opinions about end-of-life decisions. Regarding psychological distress, 67% of the respondents reported the presence of anxiety and 78.5% reported depressive thoughts. Mean QoL was −0.81, with 65 participants having a QoL score below 0 (49%), 45 a score above 0 (34%), and 24 a score of 0 (18%) (see Figure 3 for the repartition of the QoL scores). Regarding the end-of-life decision, euthanasia was never considered by about the half of the sample (49%), and most of the caregivers (48%) wanted as much therapy as possible for the patient.

### 3.2. Characteristics of Caregivers Depending on Their QoL and Psychological Distress

First, respondents were divided into two groups: positive QoL (ACSA score > 0, N = 45) and negative QoL (ACSA score < 0, N = 65). Those who had an ASCA score of 0 were not considered in these analyses (N = 24). Analyses revealed a significant difference between them concerning the time since the brain injury (H = 14.02, *p* < 0.001), with a shorter time since the brain injury associated with a negative QoL (M = 24.05 months), and a longer time associated with a positive QoL (M = 60.41 months). This result was confirmed with a significant positive correlation between QoL and time since brain injury (r = 0.38; *p* < 0.001, see Figure 4). In addition, the care setting of the patient was also different according to the QoL (X^2^ = 12.29, *p* = 0.012). More specifically, caregivers with a negative QoL more frequently had their relative in a rehabilitation center (51%), while those with a positive QoL more frequently had their relative in a nursing home (56%). Caregivers with a negative QoL also reported more unmet needs (H = 4.73; *p* = 0.030), more anxiety (X^2^ = 14.91, *p* < 0.001), and more depressive thoughts (X^2^ = 8.36, *p* = 0.015) than those with a positive QoL. The instrumental support need was also more satisfied among the caregivers who had a positive QoL (H = 6.07, *p* = 0.014; 71% vs. 56%). There was no significant difference on the other variables (i.e., gender, age of the caregiver and the patient, employment status, relationship with the patient, patients’ level of consciousness, end-of-life decisions, importance, and satisfaction of other needs) between the caregivers who had a positive vs. a negative QoL.

Second, we used the categories from the questionnaire to differentiate participants with absent, moderate, or extreme anxiety, and participants with absent, occasional, or frequent depressive thoughts. Regarding depressive thoughts, they were associated with a more frequent consideration of euthanasia (X^2^ = 18.32; *p* = 0.001). More precisely, 54% and 42% of the caregivers who currently considered euthanasia reported occasional or frequent depressive thoughts. The number of unmet needs was higher in caregivers who also reported anxiety and depressive thoughts (H = 6.33; *p* = 0.042, and H = 8.20; *p* = 0.017, respectively, see Figure 5). The importance of emotional and professional supports was also higher when the caregiver also suffered from more anxiety (H = 9.89; *p* = 0.007, and H = 8.36; *p* = 0.015, respectively), and more depressive thoughts (H = 17.48, *p* < 0.001, and H = 6.87, *p* = 0.032, respectively). In addition, the needs related to instrumental support and involvement in care were less satisfied among caregivers who had more anxiety (H = 8.02, *p* = 0.018, and H = 13.98; *p* < 0.001), and the need related to professional support was less satisfied among caregivers who had more depressive thoughts (H = 6.97; *p* = 0.031). Finally, those who had worse anxiety also had more frequent depressive thoughts (X^2^ = 35.52, *p* < 0.001).

### 3.3. Differences Linked to Other Socio-Demographic and Medical Factors

Regarding gender, the only significant difference between men and women concerned the level of importance attributed to instrumental support, with women considering it to be more important than men (H = 5.56, *p* = 0.018). The importance and satisfaction of the other needs did not differ according to the caregiver’s gender, nor did the end-of-life decisions. Regarding the relationship with the patient, it seems that the need for involvement in care was particularly important (i.e., score > 80%) among parents and spouses (H = 11.75, *p* = 0.038). Regarding the patients’ level of consciousness, it appeared that among the caregivers who considered euthanasia, 61% were relatives of patients in UWS or MCS (X^2^ = 28.46, *p* = 0.002). Opinion regarding the continuation of therapy was also different (X^2^ = 32.58, *p* = 0.037), with progressive discontinuation of therapy considered only in cases of patients in UWS and MCS. The importance and satisfaction of the needs, as well as psychological distress, did not differ according to the patients’ level of consciousness. Finally, no significant differences were noted relative to the patient’s care setting, except for the consideration of euthanasia (X^2^ = 17.70, *p* = 0.024). Indeed, among the caregivers who considered euthanasia, 73% had their relatives living in a nursing home, 15% in a rehabilitation center, and 12% at home.

Our main findings regarding the caregiver’s and patient’s characteristics that influenced the caregiver’s QoL and needs are summarized in Table 5.

## 4. Discussion

Due to progress in intensive care, more patients survived their severe brain injuries, but some may evolve towards a PDoC [32,33], leading to a high burden for their family caregiver. These caregivers report many physical, psychological, social, and environmental complaints, and a high need for involvement in care and medical information regarding their relative [8,10,11,13,14,15,17,18]. Several factors linked to the caregiver, or the patient, can influence the QoL of the caregivers, but studies are quite contradictory. We collected questionnaires from 177 family caregivers of patients with PDoC, who were mainly women (57%), and the parent of the patient (48%). Half of the patients were in an MCS (50%), with time since brain injury ranging from 1 month to nearly 30 years, and mostly hospitalized in a neuro-rehabilitation center or a nursing home (75%). These results are in line with other studies showing that the family caregivers are mainly female and that patients are usually in a long-term care facility [13,14,18,19,25,34,35].

### 4.1. Caregivers’ Needs and QOL

Regarding caregivers’ needs, our results showed that caregivers reported a lot of important or very important needs (79%). However, 44% of their needs were not entirely satisfied. This confirms previous studies which underlined the high number of important and unmet needs in family caregivers of patients with severe brain injury [8,10,15,18,19]. The most important category of needs was the need for health information, and it was also the second most satisfied, which is in line with the literature [15,18]. Thus, it seems that despite the difficulty to access the medical teams and information experienced by many caregivers [8,10], this need is generally well satisfied (mean satisfaction score of 73% in our sample). One possible explanation is that, given its importance, caregivers access medical information through different means outside the health care system (e.g., internet search, external expert opinion, social media). It is also possible that healthcare professionals are becoming increasingly aware of the need for family caregivers to receive complete information about their relative’s health and answers to their questions. The need for emotional support was scored as the least important, and the least satisfied, which is in line with some previous studies on caregivers of patients with PDoC [15,18]. This most likely does not reflect a disinterest from the caregivers towards their own emotional needs (since its mean importance is of 63% so still high) but rather reflects caregivers’ tendency to put the patient’s needs before their own. Regarding psychological distress, 67% and 79% of the respondents reported anxiety and depressive thoughts, respectively, underlining once again the high level of psychological distress in this population. Mean QoL was −0.81, corresponding to a low QoL, and only 33% of them had a positive quality of life (i.e., ACSA score > 0). These results were expected and are in line with numerous previous studies showing high psychological distress and low QoL among caregivers of patients with severe brain injury [8,11,13,14,15,17,19]. For example, in the study of Lugo et al. [15], 86% of the caregivers of LIS patients reported anxiety, 64% reported depressive thoughts, and mean QoL was −0.62. Regarding end-of-life decisions, almost half of the caregivers never considered euthanasia, and were not willing to stop the patient’s therapy. On the other hand, euthanasia was desired by 15% of the respondents. These caregivers also reported more depressive thoughts, while their relative was generally in an UWS or MCS and hospitalized in a nursing home. In comparison, the study of Lugo et al. [15] asked the same questions to caregivers of LIS patients, and 7% of them were considering euthanasia for their relatives. This difference may be due to the fact that our study mostly included UWS and MCS patients rather than LIS patients (3% of the sample). Indeed, LIS patients have the ability to choose for themselves, and these end-of-life decisions are generally not the responsibility of their caregiver [36].

### 4.2. Differences According to the Level of QoL

According to our results, those with worse QoL were relatives of patients with a shorter time since brain injury, suggesting that QoL improved with time passing. This could be due to a better acceptation of the situation and the development of more efficient coping strategies [8]. Previous studies on the subject are nevertheless quite contradictory, as some of them suggested that burden worsens, or stays the same over time [17]. A negative QoL was also more frequent among those whose relative was living in a long-term care facility (i.e., rehabilitation center or nursing home). Indeed, among caregivers with positive QoL, 24% had their relative at home, while it was the case for only 14% of those who had a negative QoL. One could have expected that having the patient at home would represent an additional burden, with a more negative impact on QoL [11]. However, a study showed that most caregivers of patients with PDoC spent most of their time with them, whether in an institution or at home, suggesting that having the relative hospitalized was not necessarily linked with a decreased burden and a better QoL [25]. In addition, patients kept at home are usually the ones with a longer time since brain injury, which we found to be positively correlated to QoL as well. Caregivers with negative QoL also reported more anxiety, more depressive thoughts, and more unmet needs as expected given the existing literature in the field [8,14,15,17]. The need for instrumental support was also more satisfied among those with a positive quality of life, suggesting that receiving help for the daily tasks and for the patient’s care, as well as paying attention to one’s own needs (e.g., sleep, time with friends, personal interests), seem important to have a better QoL in this context. No differences on gender, age of the caregiver and the patient, relationship with the patient, caregiver’s employment status, caregiver’s opinions regarding end-of-life decisions, nor patients’ level of consciousness were found between caregivers with positive vs. negative QoL, similarly to other studies on caregivers of patients with severe brain injury [12,13,15,17,21,24].

### 4.3. Differences According to the Level of Emotional Distress and Other Factors

Regarding the differences according to the level of anxiety and depressive thoughts, it appeared that caregivers with more depressive thoughts were also more likely to consider euthanasia for their relatives. Those with higher anxiety or higher depressive thoughts also reported more unmet needs, which was expected and confirms a previous study by Lugo and colleagues [15]. Emotional and professional supports were particularly important for those caregivers with high psychological distress. One explanation could be that caregivers with higher distress also have less resources to deal with their emotions, their own needs, and their relative’s needs. They may need more help at these levels than caregivers with less anxiety and depressive thoughts. Indeed, it is known that emotional distress is linked with a difficulty to mobilize personal resources and more dysfunctional coping strategies [37,38]. Our results confirmed the link between anxiety and depression as, in our sample, those who suffered from higher anxiety also reported more frequent depressive thoughts.

Other differences linked to socio-demographic and medical factors appeared. First, instrumental support was particularly important for women. This can be due to the fact that the caregiving role is generally devoted to women, in addition to their other responsibilities and workload, with a high impact on their mental and physical health [39,40]. This could lead to a higher need for help, and less time to focus on their own needs. It is also known that self-care practice, and thus focus on personal needs, is more common among female caregivers compared to males [41]. Second, involvement in care was particularly important for parents and spouses when compared to the other categories of relatives. This suggests that parents and spouses particularly want to be informed of the state of the patient, to give their opinions and know that these opinions are considered. This could be understood by the fact that parents and spouses are usually the closest relatives of the patient and thus feel particularly involved in their care. Finally, differences were also noted according to the patient’s care setting, with most of the caregivers who considered euthanasia having their relative staying in a nursing home. This may be related to the fact that patients in nursing homes, as compared to patients in a rehabilitation center, are mainly chronic patients who usually show fewer signs of improvement. Additionally, these nursing homes are often primarily designed for disabled elderly people, and hence the type of care that is provided may not correspond to the relatives’ expectations [42]. Except for this association, our results showed no differences in terms of QoL, psychological distress, or need in relation to the patient’s care setting, in line with the results of Giovannetti et al. [14] who did not find any influence of the care setting on the caregiver burden. There were no significant differences in terms of caregivers’ QoL, anxiety, depressive thoughts, or needs in relation to the patient’s level of consciousness either. Here, again, the existing literature is quite contradictory, as some studies found such differences [9,13,16,22,43], while others showed no difference in caregiver burden, psychological symptoms, and needs in relation to the patient’s diagnosis [13,17,21,24].

### 4.4. Limitations

This study suffers from several limitations. First, the cross-sectional design of the study has its own intrinsic weaknesses, as it does not allow to conclude causal relationships between our variables. Second, we did not consider other important variables such as financial issues, the strength of the relationship between the caregiver and the patient, family stressors (e.g., other ill family members), religious beliefs, and/or job stress that could have impacted the caregivers’ answers to the FNQ and the study outcome. Regarding the importance of religious beliefs in this context, it has been shown that caregivers of DoC patients often use religion as a coping strategy [19], and that a higher spirituality is associated with less burden [44]. Additionally, even if it aimed to shorten the length of the survey, the assessment of anxiety and depressive thoughts could also have been more robust by using standardized questionnaires such as the Beck Depression Inventory [45] and State-Trait Anxiety Inventory [46], instead of a single self-reported measure. Fourth, even though our sample was much larger than most studies on caregivers of patients with PDoC [14,15,22,25,47,48], the recruitment was based on a voluntary basis which might have introduced bias. Indeed, caregivers of such patients are frequently unwilling to take time away from their relatives to participate in scientific studies [17]. Caregivers who participated in this study might have had more resources and therefore might have scored differently the importance of needs and associated satisfaction levels. Finally, no standardized assessment of consciousness was used, leading to a high number of patients with unspecified diagnoses (20%) and most likely mislabeled levels of consciousness. This highlights again the importance of providing accurate medical information to the caregivers of PDoC patients.

### 4.5. Perspectives

This study introduces several scientific and clinical perspectives. First, assessing QoL, psychological distress, and the needs of caregivers of patients with severe brain injury among a larger sample and with more standardized tools would be useful. It would also be important to pay attention to the patient’s level of consciousness and care setting to equally distribute these variables in the sample, and obtain more representative and generalizable results. In addition, international multi-centric longitudinal studies would be needed to better understand the evolution of QoL, psychological distress, and needs of family caregivers of patients with PDoC, and to define the profiles of particularly at-risk caregivers. It would also be interesting to investigate how other variables might influence the caregivers’ QoL and needs, such as the prolonged grief, the attachment style, the burden, or the coping strategies [9,21,48,49,50], and to determine potential protective factors for the caregivers’ QoL. Finally, the implementation of interventions to support these caregivers is increasingly important and urgent, given their high burden and the lack of support most of them experience [10]. These could include, for example, assistance in completing forms and coordinating their relative’s care, home support services, respite care options, emotional support, and peer support [10,43,51,52,53]. Very few studies assessed the feasibility and efficacy of interventions addressed to caregivers of patients with PDoC. Among them, Li and Xu [52] randomized 107 family members of patients in UWS into one single-session psychological crisis intervention based on the development of coping abilities, and one control group. They showed that after the intervention, the participants reported lower anxiety and depression, among other symptoms. In the same way, in 2015, Corallo et al. [43] randomized 48 caregivers of patients with UWS or MCS into one intervention group consisting of 6-month long psychological support (i.e., sharing of experiences, enhancement of communication and coping skills, management of their relative) and one control group. They showed that the caregivers from the intervention group had a better QoL, less anxiety, and less depression.

## 5. Conclusions

Our study underlines the low QoL, high psychological distress, and high number of important and unmet needs among caregivers of patients with PDoC. More specifically, the need for medical information is particularly important (95%), and fortunately it seems to be quite well satisfied (73%). The need for emotional support remains high but is the least important and the least satisfied, except in the case of high emotional distress, where it becomes more important. Our results also suggest that the caregivers with lower QoL are those whose relative is hospitalized, and with shorter time since brain injury. It is however noteworthy that the care setting and time since brain injury are clearly linked, as patients with longer time since brain injury are more likely to be kept at home rather than in a specialized institution. This could explain the relationship we obtained between the care setting and the caregiver’s QoL. Those with lower QoL also report more distress and unmet needs. Even if our results must be considered with caution and need to be replicated, they underline the need to take care of the caregivers and to design interventions to address their difficulties. For example, improving emotional support in highly distressed caregivers could be of interest to increase their well-being. Further research should focus on the creation and assessment of psychosocial interventions for the caregivers of PDoC patients. Developing more initiatives which provide support groups for families, as well as accessible medical and practical information on severe brain injuries (e.g., www.mindcare.foundation (accessed on 21 November 2022)) would also be crucial. This could allow to better meet the needs that are highly valued by caregivers in the near future.

## Figures and Tables

**Figure 1 brainsci-13-00308-f001:**
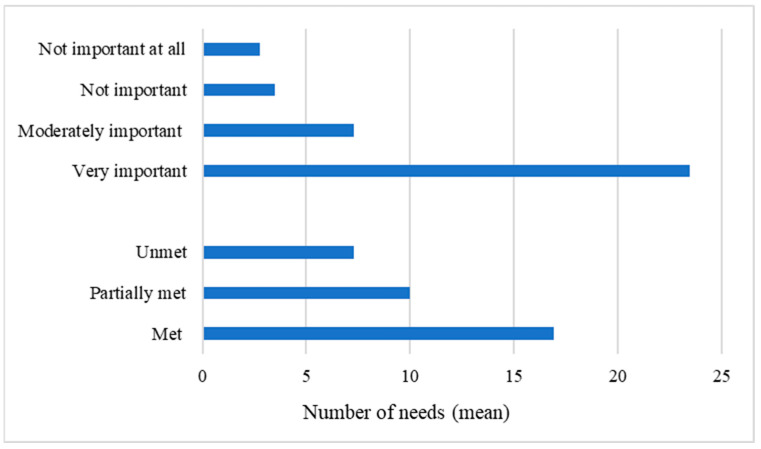
Number of needs according to their importance and satisfaction.

**Figure 2 brainsci-13-00308-f002:**
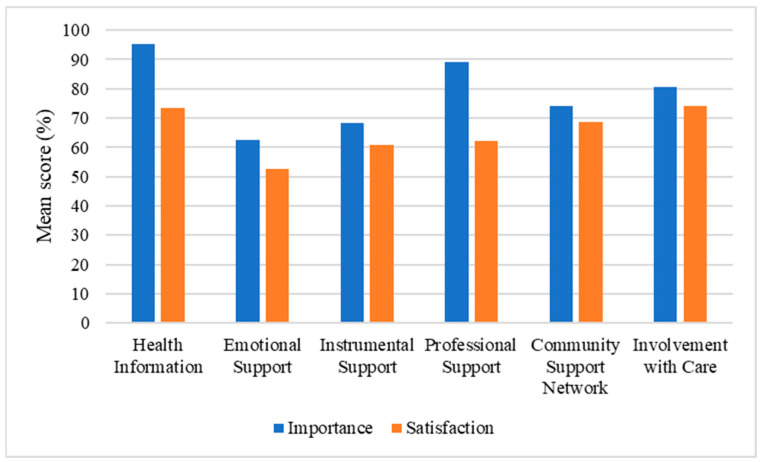
Mean importance and satisfaction for each category of need.

**Figure 3 brainsci-13-00308-f003:**
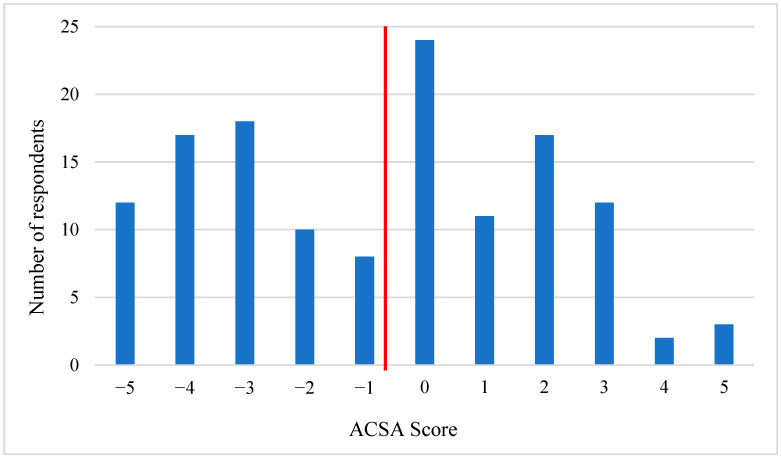
Repartition of the QoL score (ACSA). The red line corresponds to the mean QoL of the sample (−0.81).

**Figure 4 brainsci-13-00308-f004:**
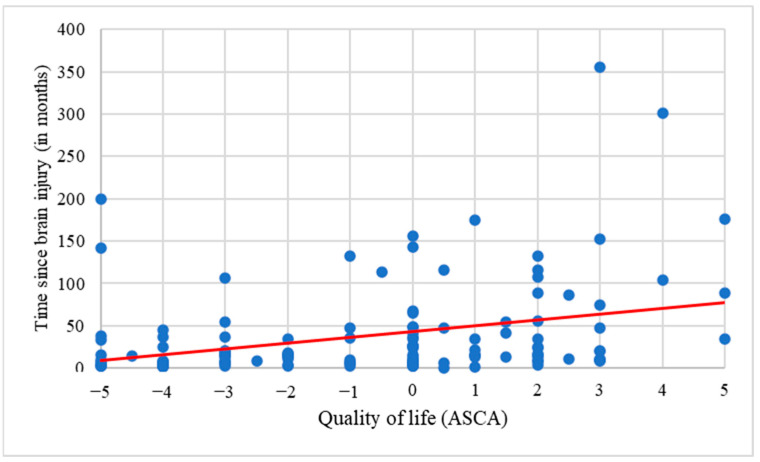
Correlation between QoL and time since brain injury (r = 0.038, *p* < 0.001).

**Figure 5 brainsci-13-00308-f005:**
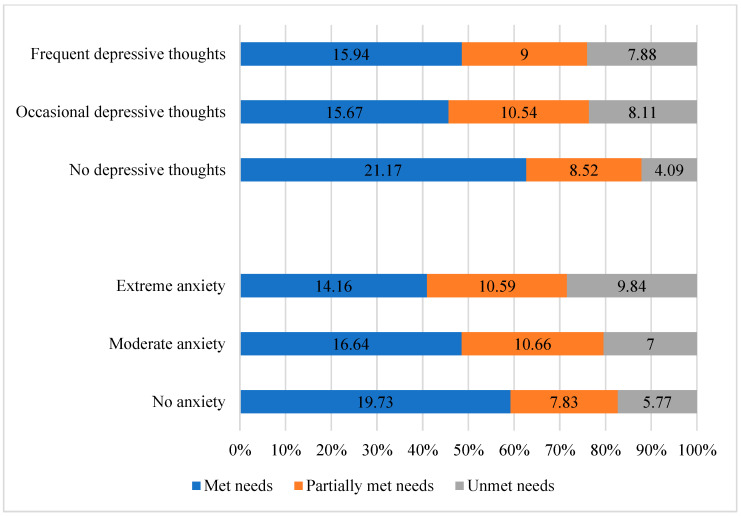
Number of met, partially met, and unmet needs according to the anxious and depressive symptoms.

**Table 1 brainsci-13-00308-t001:** Summary of previous findings regarding the impact of caregiver’s and patient’s characteristics on the caregiver’s QOL and needs.

		Impact on the Caregiver’s QoL and Needs
Characteristics linked to the caregiver	Gender	- Female: ↗ depression, ↗ anxiety, ↗ prolonged grief disorder [9,16,17,20]. - No difference in burden [21].
	Age	- Young age: ↗ prolonged grief disorder [17]. - No link between age and depression [17].
	Relationship with the patient	- Spouses: ↗ anxiety, ↗ burden [16,22]. - No difference in burden [12].
	Presence of psychological distress	- ↘ QoL [8,14,17].
	High number of unmet needs	- ↘ QoL [15].
Characteristics linked to the patient	Age	- Younger patient: ↗ prolonged grief disorder, ↗ anxiety [16,17].
	Time since brain injury	- ↗ need for social support shortly after the brain injury, and ↗ financial, social and marital negative consequences over time [10]. - ↘ QoL, ↘ number of needs, ↘ social support over time [8,17]. - Evolution of distress and burden variable [16,17]. - No link between burden and time since brain injury [13].
	Diagnosis/level of consciousness	- MCS (compared to UWS): ↗ need for help, supportive assistance [23] and emotional and social supports [13]. - Higher level of consciousness: ↘ anxiety [16]. - No differences in burden, psychological symptoms, and needs [13,17,24].
	Care setting	- ↗ or ↘ of anxiety when relative kept at home [11]. - No difference in burden, distress or grief according to the care setting [17,25]. - No influence in burden [14].

**Table 2 brainsci-13-00308-t002:** Baseline socio-demographic characteristics of the sample.

Caregivers’ Characteristics	N = 177
Age (years)	
Mean (SD) Range	52.25 (13.14) 21–86
Gender, N (%)	
Women	100 (56.5)
Men	73 (41.2)
Other	2 (1.1)
Unspecified	2 (1.1)
Relationship to the patient, N (%)	
Wife/husband/partner	58 (32.8)
Father/mother	84 (47.5)
Son/daughter	16 (9.0)
Brother/sister	15 (8.5)
Other	2 (1.1)
Unspecified	2 (1.1)
Employment status, N (%)	
Actively working	106 (59.9)
Not actively working (i.e., retired, housewife/househusband, student, unemployed)	59 (33.3)
Unspecified	12 (6.8)
Patients’ characteristics	
Age (years)	
Mean (SD)	42.69 (15.25)
Range	5–79
Level of consciousness, N (%)	
Coma	2 (1.1)
UWS	25 (14.1)
MCS	89 (50.3)
EMCS	16 (9.0)
LIS	5 (2.8)
Unspecified (no answer or multiple answers)	40 (22.6)
Time since brain injury (months)	
Mean (SD)	43.77 (64.01)
Range	1–356
Care setting, N (%)	
Home	38 (21.5)
Rehabilitation center	58 (32.8)
Nursing home	74 (41.8)
General hospital	2 (1.1)
Combination of two places	1 (0.6)
Unspecified	4 (2.3)

**Table 3 brainsci-13-00308-t003:** Caregivers’ needs importance and satisfaction (FNQ).

Need	Importance (%)	Satisfaction (%)
	Mean (SD)	Mean (SD)
Health Information	95.40 (10.77)	73.29 (25.69)
Emotional Support	62.67 (24.60)	52.76 (30.06)
Instrumental Support	68.38 (26.18)	60.75 (30.05)
Professional Support	89.19 (15.03)	62.22 (29.72)
Community Support Network	74.25 (21.76)	68.59 (26.56)
Involvement with Care	80.56 (21.22)	74.01 (28.02)

**Table 4 brainsci-13-00308-t004:** Caregivers’ quality of life and opinions about end-of-life decisions.

	N = 177
Anxiety, N (%)	
Absent	37 (20.9)
Moderate	91 (51.4)
Extreme	28 (15.8)
Unspecified	7 (4.0)
Depressive thoughts, N (%)	
Frequent	28 (15.8)
Occasional	111 (62.7)
Absent	32 (18.1)
Unspecified	6 (3.4)
Quality of life (Anamnestic Comparative Self-Assessment Scale)	
Mean (SD)	−0.81 (2.69)
Range	−5.0–5.0
Missing data (N, (%))	43 (24.3)
Opinion regarding patient’s euthanasia, N (%)	
Never considered	87 (49.2)
Considered for a while but not anymore	52 (29.4)
Desired	26 (14.7)
Unspecified	12 (6.8)
Opinion regarding patient’s therapy, N (%)	
No limit to therapy	85 (48.0)
Do not reanimate	51 (28.8)
Do not add any therapy or extend the ongoing therapy	13 (7.3)
Progressive discontinuation of therapy	6 (3.4)
Unspecified/undecided	22 (12.4)

**Table 5 brainsci-13-00308-t005:** Influence of caregiver’s and patient’s characteristics on the caregiver’s QOL and needs: summary of our findings.

		Impact on the Caregiver’s QoL and Needs
Characteristics linked to the caregiver	Gender	- Female: ↗ importance of instrumental support.
	Age	- No difference noted on all the variables.
	Relationship with the patient	- Spouses and parents: ↗ importance of involvement in care.
	Presence of psychological distress	- ↗ consideration of euthanasia. - ↗ number of unmet needs. - ↗ importance of emotional and professional supports. - ↘ satisfaction of the needs for instrumental support and involvement in care (when higher anxiety), and of the need for professional support (when more frequent depressive thoughts).
	High number of unmet needs	- ↘ QoL. - ↗ anxiety, ↗ depressive thoughts.
Characteristics linked to the patient	Age	- No difference noted on all the variables.
	Time since brain injury	- Longer time: ↗ QoL.
	Diagnosis/level of consciousness	- UWS and MCS: ↗ consideration of euthanasia and progressive discontinuation of therapy.
	Care setting	- Better QoL when patient at home or in nursing home compared to rehabilitation center. - Nursing home: ↗ consideration of euthanasia.

## Data Availability

The dataset of this study is available upon reasonable request by contacting the corresponding author.

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
