# Peer review of "Needs and Quality of Life of Caregivers of Patients with Prolonged Disorders of Consciousness"

_brainsci, 2023, doi:10.3390/brainsci13020308_

Round 1

Reviewer 1 Report

I enjoyed reviewing this article very much.  I think it will make a great contribution to the literature on this important topic. 

Recommendations for editing are limited to minor grammatical errors/typos noted:

-       Line 19 - “Many patients with severe brain damages may survived…”

-       Line 49 - “Coma, UWS and MCS are often refer under the umbrella of…”

-    Line 223 – the title for Table 2 should be moved to the next page so its on the same page with the table

Author Response

We thank the reviewers for their time and their comments regarding our manuscript. Below, you will find our responses 
point-by-point to their suggestions.
Reviewer #1:
- Line 19, the following sentence has been corrected: “Many patients with severe brain damage may survive”.
- Line 49, the following sentence has been corrected: “Coma, UWS and MCS are often referred under the 
umbrella of “disorders of consciousness” (DoC).”
- The title of Table 2 is now on the same page than the table.
Reviewer #2: Similar surveys were conducted among relatives of critically ill patients in ICU during the formation of 
the end-of-life management protocol. In this case, for obvious reasons, religious affiliation of the family was taken into 
consideration. It would be interesting to know the authors opinion on how important this aspect is within the framework 
of DoC caregivers QoL.
âž” The interest of considering religious beliefs in this context has been cited line 437. We added the following 
sentence in order to discuss that point with more details (line 438): “Regarding the importance of religious beliefs 
in this context, it has been shown that caregivers of DoC patients often use religion as a coping strategy [19],
and that a higher spirituality is associated with less burden [44].”
Reviewer #3: Some details could be made clearer, such as the approaches used to diagnose the patients (it is discussed 
in the Limitations section that there was no standardised assessment of consciousness) and the geographical region for 
a better understanding of the types of care facilities. However, these details do not affect the conclusions and the overall 
value of the work.
âž” More detail about the diagnosis assessment has been given Line 161: “Regarding the patients’ level of 
consciousness, no standardized assessment has been done, and this information was given by the caregiver.”
âž” More detail about the geographical region has been given Line 145: “the patient was living at home or 
hospitalized in one of the subacute or long-term care facility involved in the study [26], mostly in Belgium but 
also outside Belgium (N=32; France, Holland, Germany, Luxembourg, Italy, United-Kingdom or Greece).”
Additional modification :
- The title has been corrected (“disorders” instead of “disorder”).
- The formatting of Tables 1 and 5 have been changed.

Reviewer 2 Report

Dear colleagues. I would like to thank the authors for the extremely interesting and important work they have done. The  raised topic is of great importance, since a severe and prolonged illness of a family member always affects the life of the family in all aspects.

Similar surveys were conducted among relatives of critically ill patients in ICU during the formation of the end-of-life management protocol. In this case, for obvious reasons, religious affiliation of the family was taken into consideration. It would be interesting to know the authors opinion on how important this aspect is within the framework of DoC caregivers QoL.

Author Response

(The authors gave the same response as above.)

Reviewer 3 Report

The study by Dr Gosseries et al. provides a comprehensive analysis of the needs, quality of life and distress of 177 primary caregivers of patients with disorders of consciousness (mostly of the prolonged type, of which MCS accounted for half the cases). The hypotheses, data collection methods and results are robust and consistent. Although some conclusions may seem intuitively clear, the study provides reliable insights into the impact of severe and prolonged brain damage on the life of the caregiver. The paper is also valuable for the directions it gives to further improve care for the patient's relatives. 

Some details could be made clearer, such as the approaches used to diagnose the patients (it is discussed in the Limitations section that there was no standardised assessment of consciousness) and the geographical region for a better understanding of the types of care facilities. However, these details do not affect the conclusions and the overall value of the work.

Author Response

(The authors gave the same response as above.)
